# The Mechanism of Interleukin 33-Induced Stimulation of Interleukin 6 in MLO-Y4 Cells

**DOI:** 10.3390/ijms241914842

**Published:** 2023-10-02

**Authors:** Sae Noguchi, Ryota Yamasaki, Yoshie Nagai-Yoshioka, Tsuyoshi Sato, Kayoko Kuroishi, Kaori Gunjigake, Wataru Ariyoshi, Tatsuo Kawamoto

**Affiliations:** 1Division of Orofacial Functions and Orthodontics, Kyushu Dental University, 2-6-1 Manazuru, Kokurakita-ku, Kitakyushu, Fukuoka 803-8580, Japan; r20noguchi@fa.kyu-dent.ac.jp (S.N.); kayo-na@kyu-dent.ac.jp (K.K.); k-kaori@kyu-dent.ac.jp (K.G.); r15kawamoto@fa.kyu-dent.ac.jp (T.K.); 2Division of Infections and Molecular Biology, Department of Health Promotion, Kyushu Dental University, 2-6-1 Manazuru, Kokurakita-ku, Kitakyushu, Fukuoka 803-8580, Japan; r18yamasaki@fa.kyu-dent.ac.jp (R.Y.); r16yoshioka@fa.kyu-dent.ac.jp (Y.N.-Y.); 3Department of Oral and Maxillofacial Surgery, Saitama Medical University, 38 Moro-hongou, Moroyama-machi, Iruma-gun, Saitama 350-0495, Japan; tsato@saitama-med.ac.jp

**Keywords:** osteocyte, IL-33, IL-6, bone remodeling, MAPK, NF-κB, AP-1

## Abstract

The differentiation and function of osteocytes are controlled by surrounding cells and mechanical stress; however, the detailed mechanisms are unknown. Recent findings suggest that IL-33 is highly expressed in periodontal tissues in orthodontic tooth movement. The present study aimed to elucidate the effect of IL-33 on the expression of regulatory factors for bone remodeling and their molecular mechanisms in the osteocyte-like cell line MLO-Y4. MLO-Y4 cells were treated with IL-33, and the activation of intracellular signaling molecules and transcriptional factors was determined using Western blot analysis and chromatin immunoprecipitation assay. IL-33 treatment enhanced the expression of IL-6 in MLO-Y4 cells, which was suppressed by the knockdown of the IL-33 receptor ST2L. Additionally, IL-33 treatment induced activation of NF-κB, JNK/AP-1, and p38 MAPK signaling pathways in MLO-Y4 cells. Moreover, pretreatment with specific inhibitors of NF-κB, p38 MAPK, and JNK/AP-1 attenuated the IL-33-induced expression of IL-6. Furthermore, chromatin immunoprecipitation indicated that IL-33 increased c-Jun recruitment to the IL-6 promoter. Overall, these results suggest that IL-33 induces IL-6 expression and regulates osteocyte function via activation of the NF-κB, JNK/AP-1, and p38 MAPK pathways through interaction with ST2L receptors on the plasma membrane.

## 1. Introduction

In orthodontic tooth movement (OTM), osteocytes in alveolar bone and periodontal ligament cells respond to mechanical stress [1], and these cellular responses are regulated by cytokines, such as interleukin 1 beta (IL-1β) and interleukin 6 (IL-6), released from several cells. Periodontal tissue undergoing OTM is divided into the compression and tension sides by orthodontic force. Bone resorption is induced on the compression side due to increased expression of receptor activator of NF-κB ligand (RANKL), but inhibited on the tension side due to increased expression of osteoprotegerin (OPG) [2]. The binding of RANKL to RANK in osteoclast progenitor cells promotes osteoclast differentiation. Moreover, OPG secreted from osteoblasts and osteocytes binds to RANKL and inhibits excessive bone resorption by evading RANKL–RANK interaction [3].

Osteocytes account for 90–95% of all bone constituent cells in the adult skeleton. During the mineralization of the osteoid ma trix, osteoblasts are embedded in the mineral phase and differentiate into osteocyte bodies. Osteocytes are star-shaped cells with cytoplasmic processes and cilia (approximately 40–100 per cell), and are smaller than osteoblasts. Osteocytes are distributed throughout the bone and are interconnected with each other and with other bone cells. Surrounded by bone fluid in the bone lacuna, osteocytes can sense physical changes caused by mechanical loading on the bone and express several factors that regulate bone formation or resorption [4,5]. This mechanism is known as “mechanotransduction” and is regulated by several cytokines and hormones, among which interleukin 6 (IL-6) is a multifunctional cytokine that controls regeneration, metabolism, homeostasis of bone tissue, and the immune system [6]. IL-6 acts on osteocytes, osteoblasts, and synovial fibroblasts to promote RANKL expression and osteoclast differentiation [7,8,9]. Additionally, RANKL suppresses osteoblast differentiation by inhibiting the expression of runt-related transcription factor 2 (Runx2) and osterix in osteoblast progenitor cells [10] and by inhibiting the Wnt signaling pathway in mesenchymal precursor cells [11].

IL-33 is a cytokine that belongs to the IL-1 family members [12]. The IL-33 receptor is a heterodimeric complex consisting of suppression of tumorigenicity 2 (ST2, also designated ST2L), which is expressed on the surface of various types of cells, and IL-1 receptor accessory protein (IL-1RAcP) [13]. IL-33 shows important roles as nuclear alarmins that initiate immune responses during infection-, physical damage-, or chemical damage-induced necrosis [12]. A previous study showed that IL-33 and ST2L expression levels were enhanced by mechanical stimulus in the periodontium of BALB/c mice and in MC3T3-E1 osteoblastic cells [14]. Additionally, mechanical loading reduced osteoblast numbers and decreased mRNA expression of osteoblast markers in the periodontium of ST2-deficient mice [14]. Moreover, IL-33 promoted matrix mineral deposition in primary osteoblasts following long-term ascorbate, dexamethasone, and β-glycerophosphate treatment [15]. In contrast, IL-33 induced osteoclast apoptosis and inhibited osteoclastogenesis in the presence of RANKL in bone marrow-derived osteoclasts [14]. Similarly, IL-33/ST2 signaling inhibits osteoclast formation and reduces bone loss by decreasing the expression of nuclear factor of activated T cells and cytoplasmic 1 (NFATc1) in osteoclast precursors [16,17]. Despite these interesting findings, the effects of IL-33 on osteocytes have not been well studied.

Presently, detailed information on the functions of osteocytes remains unclear owing to the lack of available osteocyte cell lines adaptable for in vitro experiments. The MLO-Y4 (for murine long bone osteocyte Y4) cell line has been isolated from the long bones of transgenic mice, in which the SV40 large T-antigen oncogene is expressed under the control of an osteocalcin promoter [18]. Similarly to primary osteocytes, this cell line forms extensive dendritic processes and produces large amounts of osteocalcin, but low amounts of alkaline phosphatase. Additionally, MLO-Y4 cells are positive for CD44, osteopontin, T antigen, and connexin 43. Although osteocytes are important for bone remodeling, their detailed functions are yet to be elucidated. Therefore, the present study aimed to elucidate the effect of IL-33 on the expression of regulatory factors for bone remodeling and its underling molecular mechanisms in MLO-Y4 cells.

## 2. Results

### 2.1. IL-33 Increases IL-6 Expression in MLO-Y4 Cells

Real-time RT-qPCR was performed to elucidate the effect of IL-33 on the expression of regulatory factors. IL-33 treatment upregulated *Il-6* mRNA expression in MLO-Y4 cells in both a time- and dose-dependent manner, with the expression level peaking at 50 ng/mL and 6 h after treatment, respectively (Figure 1A,B). However, IL-33 administration did not significantly affect the expression of other inflammatory cytokines, such as *Tnf-α* or *Il-1β* (below the detectable level). Additionally, IL-33 treatment slightly stimulated the expression of the osteoclastogenesis regulatory factor *Tnfsf11* (gene encoding RANKL [19]), but not *Opg*. Moreover, ELISA demonstrated that IL-33 increased IL-6 secretion in MLO-Y4 cells in a time-dependent manner (Figure 1C).

### 2.2. ST2L on the Cell Surface of MLO-Y4 Cells Is Involved in the Induction of IL-6 Expression by IL-33

Immunofluorescence staining showed that ST2L, a specific receptor for IL-33, was present on the surface of MLO-Y4 cells (Figure 2A). A member of IL-1 receptor superfamily, ST2L is a product of the *Il1rl1* gene [20]. To determine whether ST2L was involved in IL-33-induced increase in IL-6 expression, specific siRNA for *Il1rl1* was introduced by electroporation. A 67% knockdown of *Il1rl1* mRNA expression was observed in *Il1rl1* siRNA-transfected MLO-Y4 cells compared with those of cells transfected with control siRNA (Figure 2B). Moreover, IL-33-induced increase in *Il-6* mRNA expression was suppressed in MLO-Y4 cells transfected with *Il1rl1* siRNA (Figure 2C).

### 2.3. IL-33 Induces IL-6 Production via the NF-κB Signaling Pathway

Furthermore, we examined the involvement of the nuclear factor-κB (NF-κB) pathway in IL-33-induced increase in IL-6 expression. NF-κB, a transcriptional factor that regulates the expression of genes involved in inflammatory and immune responses, is composed of p65 and p50 heterodimers. The activation of NF-κB is induced by degradation of Iκ-Bα, which allows NF-κB heterodimers to migrate into the nucleus and bind to promoters of specific genes. IL-33 induced the transient degradation of Iκ-Bα protein (Figure 3A) and nuclear translocation of p65 protein after 15 min of treatment (Figure 3B). However, pretreatment of MLO-Y4 cells with BAY11-7082, which inhibits Iκ-Bα phosphorylation, suppressed IL-33-induced degradation of Iκ-Bα (Figure 3C). Additionally, pretreatment with selective inhibitors of Iκ-Bα significantly downregulated IL-33-induced increased *Il-6* mRNA expression (Figure 3D).

### 2.4. IL-33 Induces IL-6 Production via JNK and p38 MAPK-Mediated Signaling Pathways

To elucidate the intracellular signaling responsible for IL-33-induced increase in IL-6 expression, we focused on the mitogen-activated protein kinase (MAPK) signaling cascade. MAPKs are activated by phosphorylation and regulate a variety of cellular functions, including inflammatory response. At least three MAPK pathways—c-Jun N-terminal kinase (JNK), p38 MAPK, and extracellular signal-regulated kinase (ERK)-mediated signaling pathways—have been identified. IL-33 treatment stimulated the phosphorylation of JNK and p38 MAPK protein, but did not significantly affect ERK phosphorylation (Figure 4A,B). However, pretreatment with specific inhibitors of each protein effectively reduced IL-33-induced phosphorylation of JNK (Figure 5A) and p38 MAPK (Figure 5B). Additionally, IL-33-induced increase in *Il-6* mRNA expression was significantly suppressed by both JNK (Figure 5C) and p38 MAPK (Figure 5D) inhibitors. Moreover, knockdown of *Il1rl1* by siRNA inhibited IL-33-induced JNK phosphorylation, but not p38 MAPK phosphorylation (Figure 5E). Overall, these results suggest that IL-33 increases IL-6 expression via JNK and p38 MAPK-mediated signaling pathways.

### 2.5. IL-33 Induces IL-6 Production via c-Jun-Mediated AP-1 Activation

To elucidate the exact component of JNK signaling involved in IL-33-induced increase in IL-6 expression, we examined the effect of IL-33 on the phosphorylation of c-Jun, a component of AP-1 downstream of the JNK signaling pathway. Western blotting revealed that IL-33 treatment transiently enhanced c-Jun phosphorylation at 15 min of stimulation (Figure 6A). However, pretreatment with SR11302, an inhibitor of the transcriptional activity of AP-1 in vitro and in vivo [21], suppressed IL-33-induced mRNA expression of *Il-6* (Figure 6B). Moreover, chromatin immunoprecipitation (ChIP) assay showed that IL-33 treatment promoted the recruitment c-Jun protein to the AP-1 binding site in the *Il-6* promoter region in MLO-Y4 cells (Figure 6C).

## 3. Discussion

Although various studies have been conducted on the biological activity of IL-33 in immune and inflammatory responses [22], studies on its role in bone metabolism, especially in osteocytes, are limited. Immune and inflammatory responses are closely in volved in various processes, including bone metabolism and the tumor microenvironment [23]. In the present study, we evaluated the biological effects of IL-33 on osteocytes using the osteocyte-like cell line MLO-Y4. IL-33 treatment significantly upregulated IL-6 expression in a time- and concentration-dependent manner. A previous study showed that IL-33 treatment at 10 ng/mL for 24 h induced the production of inflammatory cytokines, including TNF, IL-1β, and IL-6, in dendritic cells [24]. These inflammatory cytokines have also been reported to modulate the production of hormones associated with bone metabolism, such as insulin-like growth factors [25]. Interestingly, IL-33 had no significant effects on the gene expression of other inflammatory cytokines, such as TNF-α and IL-1β, in MLO-Y4 cells. The responsiveness of cytokines to IL-33 treatment may vary among cell types. Additionally, IL-33 treatment slightly upregulated the expression of RANKL, a regulator of osteoclast differentiation, in MLO-Y4 cells, but did not affect OPG expression, which was consistent with previous findings that IL-33 stimulation induced RANKL expression but not OPG expression in osteoblasts and cementoblasts [26,27]. These results suggest that IL-33 is involved in bone remodeling process by regulating the production of various cytokines in osteocytes. Further analyses were performed to elucidate the mechanism through which IL-33 treatment induced IL-6 expression in MLO-Y4 cells.

Immunostaining confirmed that ST2L, a receptor for IL-33, is present on the cell surface of MLO-Y4. Moreover, ST2L knockdown using a specific siRNA suppressed IL-33-induced IL-6 expression, indicating that ST2L was the main receptor through which IL-33 induced IL-6 expression in osteocytes. Similarly, IL-33 promoted IL-6 secretion via ST2L in esophageal adenocarcinoma cells (OE19 and OE33) [28]. Moreover, IL-33 binding to ST2L induces the activation of various intracellular signaling molecules, including NF-κB, MAPK, nuclear factor of activated T cells (NFAT), mechanistic target of rapamycin (mTOR), signal transducer and activator of transcription 3 (STAT3), and AP-1 [29,30,31].

Previous studies have also shown that activation of the NF-κB signaling pathway is involved in the induction of IL-6 production in macrophages [32]. NF-κB/Rel proteins include p52, p50, c-Rel, RelA/p65, and RelB, which function as dimeric transcription factors and regulate the expression of several genes involved in a wide range of biological processes. The activation of the NF-κB signaling pathway induces Iκ-Bα degradation by the ubiquitin–proteasome system. During the process, the nuclear transition signal is exposed and NF-κB dimers migrate into the nucleus to initiate the transcription of various genes [33]. In the present study, IL-33 stimulation enhanced Iκ-Bα degradation and the nuclear translocation of NF-κB in osteocytes. However, suppression of Iκ-Bα protein degradation by selective inhibitors restored IL-33-induced IL-6 expression. Overall, these results indicate that IL-33-induced IL-6 expression in osteocytes is mediated by the activation of the NF-κB signaling pathway, which is consistent with previous findings on bone marrow-derived mast cells [30].

MAPK signaling is an important pathway in the regulation of growth, proliferation, differentiation, migration, and apoptosis in eukaryotic cells. Three major components of MAPK cascades—ERK1/2, JNK, and p38 MAPK—are responsible for the activation of transcription factors in the cytoplasm or nucleus, which results in the induction of expression of several genes [34]. MAPK signaling activation is involved in the induction of IL-6 expression in medullary nucleus cells [35]. Therefore, we investigated the role of the MAPK signaling pathway in IL-33-stimulated IL-6 expression in osteocytes. IL-33 treatment enhanced JNK and p38 MAPK phosphorylation, but did not affect ERK1/2 phosphorylation in MLO-Y4 cells. Additionally, inhibition of JNK and p38 MAPK activation by chemical inhibitors significantly suppressed IL-33-induced increase in IL-6 expression. These results indicate that the JNK and p38 MAPK-mediated MAPK activation is involved in IL-33-induced expression of IL-6 in osteocytes, which is consistent with previous findings on bone marrow-derived mast cells [30]. However, ST2L knockdown significantly suppressed IL-33-induced increase in JNK phosphorylation in MLO-Y4 cells, but did not affect p38 MAPK phosphorylation. This result was partially inconsistent with that of a previous study, which showed that ST2L knockdown restored IL-33-induced hyperphosphorylation of JNK and p38 MAPK in gastric cancer cells [36]. This could be due to the possibility that p38 MAPK phosphorylation in MLO-Y4 cells was sufficiently induced by the binding of IL-33 to ST2L, which was not knocked down. Additionally, interaction of IL-33 with unknown receptors other than ST2L is also speculated.

JNK is a stress-induced protein kinase that phosphorylates the transcription factor c-Jun [34]. The transcription factor AP-1 is often defined as a dimeric assembly composed of members of the Jun (c-Jun, JunB, JunD) and Fos (c-Fos, FosB, Fra-1, Fra-2) multigene families [37], and its activation is regulated by MAPK, such as ERK, JNK, and p38 MAPK [38]. Several extracellular stimuli (UV irradiation, stress, cytokines, growth factors) activate c-Jun via JNK phosphorylation and induce dimerization of AP-1 components, which bind to the TRE element of the promoter and activate gene transcription [39]. Multiple regulatory elements have been identified in the *Il-6* promoter [40]. The mouse *Il-6* promoter is predicted to have a binding site for AP-1 between −249 to −242 and a binding site for NF-κB between −45 and −35 [41]. In the present study, IL-33 treatment induced c-Jun protein phosphorylation and recruitment to the AP-1 binding site in the IL-6 promoter region in MLO-Y4 cells. Overall, these results indicate that the JNK-dependent transactivation of AP-1, as well as NF-κB activation, play a role in IL-33-induced expression of IL-6 in osteocytes. Moreover, the inhibition of AP-1 activation recovered IL-33-induced *Il-6* mRNA expression in MLO-Y4 cells, and a previous study showed that IL-33 enhanced c-Jun phosphorylation and the DNA binding activity of AP-1 in human umbilical vein endothelial cells (HUVECs) [42].

In addition to its direct effect on progenitor cells, IL-6 can indirectly promote osteoclast formation and bone resorption via stimulation of RANKL expression in osteoclast-supporting cells, such as osteoblasts [43]. In contrast, some studies have shown that IL-33 acts directly on progenitor cells to inhibit osteoclast formation and activation [14,16]. Moreover, IL-33 may positively regulate osteoblast differentiation and activation [15]. Surprisingly, in the present study, IL-33 stimulation induced *Tnfsf11* mRNA expression prior to *Il-6* in MLO-Y4 cells; therefore, further studies are necessary to elucidate the interaction of IL-33 and IL-6 in other bone component cells, such as osteoclasts and osteoblasts. Osteocytes are also considered to be one of the IL-33 producing cells [29]. Although IL-33 is highly expressed on the compression side of the periodontal ligament and its secretion is enhanced by sustained compressive force [44], the effect of orthodontic force loading on IL-33 expression in bone-forming cells is yet to be fully elucidated. Culture systems using the uniform compression method [45,46], which mimics the compression side of periodontal tissue in vitro, and orthodontic model mice [44] are necessary to elucidate the expression of IL-33 in mechanically loaded osteocytes.

Conclusively, the results suggest that IL-33 induces IL-6 expression in MLO-Y4 cells via activation of the NF-κB, JNK/AP-1, and p38 MAPK pathways by binding ST2L receptors on the cell surface, resulting in the regulation of bone component cells behaviors and homeostasis of bone tissue [4]. This study is limited to in vitro validation using osteocyte-like cell lineage and lacks analysis of primary culture and in vivo systems. Furthermore, the present study has elucidated the detailed molecular mechanisms underlying the expression and production of IL-6 by the interaction of IL-33 and ST2L in osteocyte-like cells, but the effect of induced IL-6 production on the function of osteocytes and surrounding bone component cells was not clarified. Additionally, these findings contribute to understanding of the biological function of IL-33 in bone metabolisms and the molecular mechanisms of bone remodeling.

## 4. Materials and Methods

### 4.1. Reagents and Antibodies

Recombinant human IL-33 and inhibitor of AP-1 (SR11302) were obtained from R&D Systems (Minneapolis, MN, USA). Anti-ST2 polyclonal antibody was obtained from Santa Cruz Biotechnology (Santa Cruz, CA, USA). Anti-phospho-ERK1/2 monoclonal (D13.14.4E), anti-ERK1/2 monoclonal (137F5), anti-phospho-p38 MAPK monoclonal (D3F9), anti-p38 MAPK polyclonal, anti-phospho-JNK monoclonal (81E11), anti-JNK polyclonal, anti-phospho-c-Fos monoclonal (D82C12), anti-c-Fos monoclonal (9F6), anti-phospho-c-Jun monoclonal (54B3), anti-c-Jun monoclonal (60A8), anti-IκB-α monoclonal (L35A5), and anti-NF-κB p65 monoclonal (D14E12) antibodies were purchased from Cell Signaling Technology (Beverly, MA, USA). Anti-β-actin monoclonal antibody (AC-15) and anti-TATA binding protein (TBP) polyclonal antibody were purchased from Sigma-Aldrich (St. Louis, MO, USA) and Proteintech Group (Rosemont, IL, USA), respectively. Specific inhibitors of p38 MAPK (SB203580), JNK (SP600125), and NF-κB (BAY11-7982) were purchased from Merck Millipore (Darmstadt, Germany).

### 4.2. Cell Culture

Mouse osteocyte-like MLO-Y4 cells were kindly provided by Dr. Lynda F. Bonewald (University of Missouri). MLO-Y4 cells were cultured to 70–80% confluency in a 100 mm/collagen type I-coated dish (IWAKI, Osaka, Japan) containing alpha-minimal essential medium (α-MEM; Fujifilm Wako Pure Chemical Co, Osaka, Japan) supplemented with 5% fetal bovine serum (FBS; Sigma Aldrich) and 1% penicillin–streptomycin (FUJIFILM Wako Pure Chemical Co) at 37 °C in a 5% CO_2_ humidified atmosphere. For molecular biological analysis, cells were seeded Corning^®^ BioCoat^®^ collagen I-coated plates (Corning, Corning, NY, USA) at 5.0 × 10^5^ cells/well and stimulated with IL-33 for indicated times. In some experiments, cells were pretreated for 30 or 60 min in complete medium with inhibitors.

### 4.3. RNA Extraction and Real-Time Quantitative PCR (Real-Time RT-qPCR)

Total RNA was extracted from treated cells using a Cica Geneus^®^ RNA prep kit (Kanto Chemical Co. Inc., Tokyo, Japan). cDNA synthesis by reverse transcription and quantification of mRNA by real-time RT-qPCR were carried out as described previously [47]. Relative expression levels of each mRNA were quantified by the 2∆Ct method with *Gapdh* as the housekeeping gene. The primer sequences of the target genes are indicated in Table 1.

### 4.4. Protein Extraction and Western Blotting

Proteins were extracted from the treated cells using cell lysis buffer (Cell Signaling Technology) containing a protease-inhibitor mixture (Thermo Scientific, Rockford, IL, USA) and phosphatase-inhibitor mixture (Nacalai Tesque, Kyoto, Japan). In some experiments, the nuclear and cytoplasmic fractions were extracted using NE-PER (Thermo Fisher Scientific) according to the manufacturer’s instructions. Protein concentrations were measured using a DC protein assay kit (Bio-Rad, Hercules, CA, USA). Equal volumes of protein sample were loaded on precast 12.5% polyacrylamide gel (SuperSepTM Ace; Fujifilm Wako Pure Chemical Co) and electrophoresed in Tris–glycine sodium dodecyl sulfate (SDS) running buffer. The subsequent analyses (blotting, antibody reaction and band detection) were conducted following previously reported protocols [47] with primary antibodies against Iκ-Bα, p65, phospho-JNK, JNK, phospho-p38 MAPK, p38 MAPK, phospho-ERK, ERK, phopho c-Jun, c-Jun and β-actin. The band intensity of each blot was quantified by densitometric analysis using Image LabTM^®^ 2.0 software (Bio-Rad).

### 4.5. Enzyme-Linked Immunosorbent Assay (ELISA)

Cells (5 × 10^5^ cells/well) were seeded into Corning^®^ BioCoat^®^ collagen I-coated plates containing α-MEM supplemented with 10% FBS, and treated with IL-33 (10 ng/mL) for 12–48 h. The conditioned media of treated cells were collected and clarified by centrifugation at 1200 rpm for 10 min at 4 °C. Finally, the concentration of IL-6 in the conditioned media was measured using a mouse IL-6 sandwich ELISA kit (Proteintech Group, Rosemont, IL, USA) according to the manufacturer’s instructions.

### 4.6. Immunocytochemistry

Cells were cultured in 4-well Lab-TekTM^®^ PermanoxTM chamber slides (Thermo Fisher Scientific) at a density of 1 × 10^4^ cells/well. Immunofluorescent staining was performed following a previous reported protocol [48] with primary antibody against ST2 (1:200). Alexa Fluor^®^ 488-conjugated donkey anti-goat IgG (Thermo Fisher Scientific) was used as a secondary antibody (1:200). Images were visualized by a BZ-9000 microscope (Keyence, Osaka, Japan) and processed using BZ-II imaging software (v1.1, Keyence).

### 4.7. Silencing of ST2L by Specific Small Interfering RNAs (siRNA)

siRNAs were used to knock down ST2L expression in MLO-Y4 cells. Control siRNA (1027280) and siRNA against mouse *Il1rl1* (SI02670598) were purchased from Santa Cruz Biotechnology and Qiagen (Valencia, CA, USA), respectively. The siRNAs were delivered into the cells using an NEPA21 Super Electroporator (Nepa Gene Co., Ltd., Chiba, Japan) according to the manufacturer’s instructions. Two pulses with a voltage of 125 V/20 V and a width of 7.5 ms/50 ms were applied. Transfected cells were immediately suspended with prewarmed α-MEM with 10% FBS and cultured in 6-well plates for 48 h before stimulation with IL-33 (10 ng/mL).

### 4.8. ChIP Assay

MLO-Y4 cells (1 × 10^7^/well) were seeded in 150 mm collagen type I-coated dishes (IWAKI) and incubated with IL-33 (10 ng/mL) for the indicated times. ChIP assay was performed according to the previously described procedures [48]. Antibodies specific for c-Jun were used for immunoprecipitations of chromatin fractions. The purified immunoprecipitated DNA was amplified by real-time RT-qPCR using SYBR^®^ green and normalized against the input chromatin. Primers were generated for the −350 locations relative to the transcription start sites in the mouse *Il-6* promoter: forward 5′-CCC ATC AAG ACA TGC TCA AGT G-3′ and reverse 5′-CAG GGC CTT GAT CTG TCT TC TGT GAC GTC GTT TAG CAT CG -3′. Data represent the percentage input of independent triplicate samples.

### 4.9. Statistical Analysis

Each experiment was performed a minimum of three times. Data were analyzed using Excel (Microsoft, Redmond, WA, USA) and EZR software [49] version 1.54 and a modified version of R Commander designed to add statistical functions frequently used in biostatistics. Data are expressed as means ± standard deviations (SDs), and statistical differences were determined using one-way analysis of variance (ANOVA), followed by Tukey’s or Dunnett’s post hoc test. Statistical significance was set at *p* < 0.05.

## Figures and Tables

**Figure 1 ijms-24-14842-f001:**
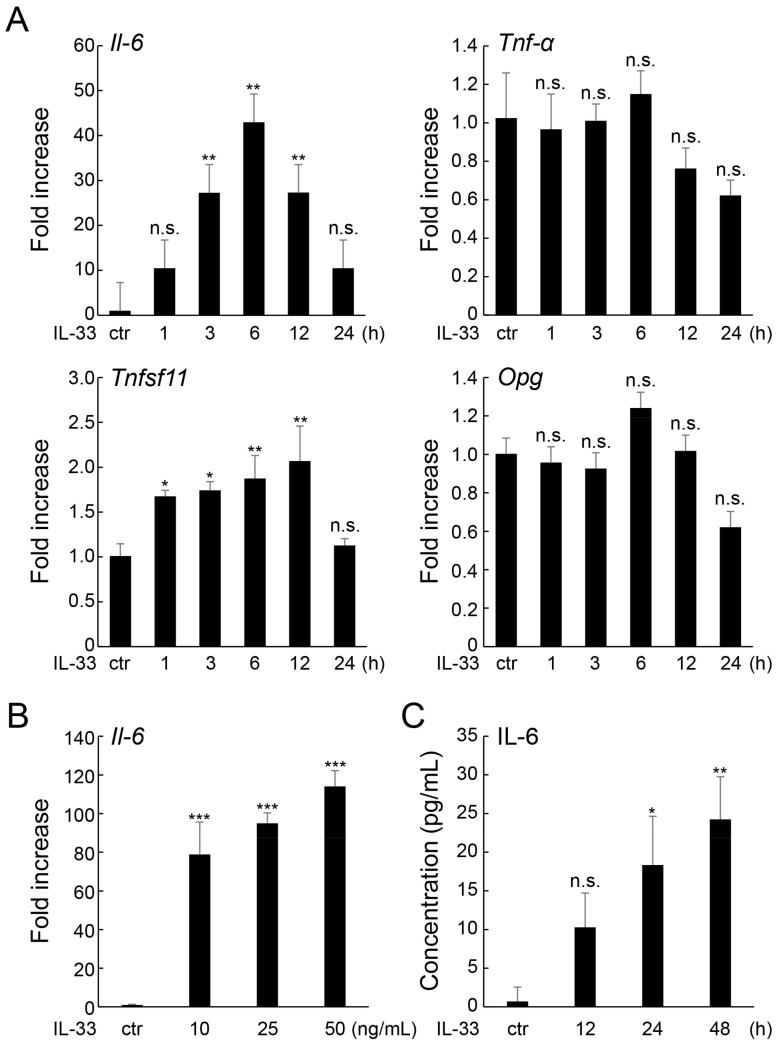
Effect of IL-33 on IL-6 expression in MLO-Y4 cells. (**A**) MLO-Y4 cells were incubated with IL-33 (10 ng/mL) for 1–24 h, and *Il-6*, *Tnf-α*, *Tnfsf11*, and *Opg* mRNA expression levels were measured using real-time RT-qPCR. (**B**) MLO-Y4 cells were incubated with IL-33 (10–50 ng/mL) for 6 h, and *Il-6* mRNA levels were measured using real-time RT-qPCR. (**C**) MLO-Y4 cells were incubated with IL-33 (10 ng/mL) for 12–24 h, and IL-6 protein levels in conditioned medium were measured using ELISA. * *p* < 0.05, ** *p* < 0.01, *** *p* < 0.001, n.s.; not significant compared with ctr (Dunnett’s post hoc test after one-way ANOVA).

**Figure 2 ijms-24-14842-f002:**
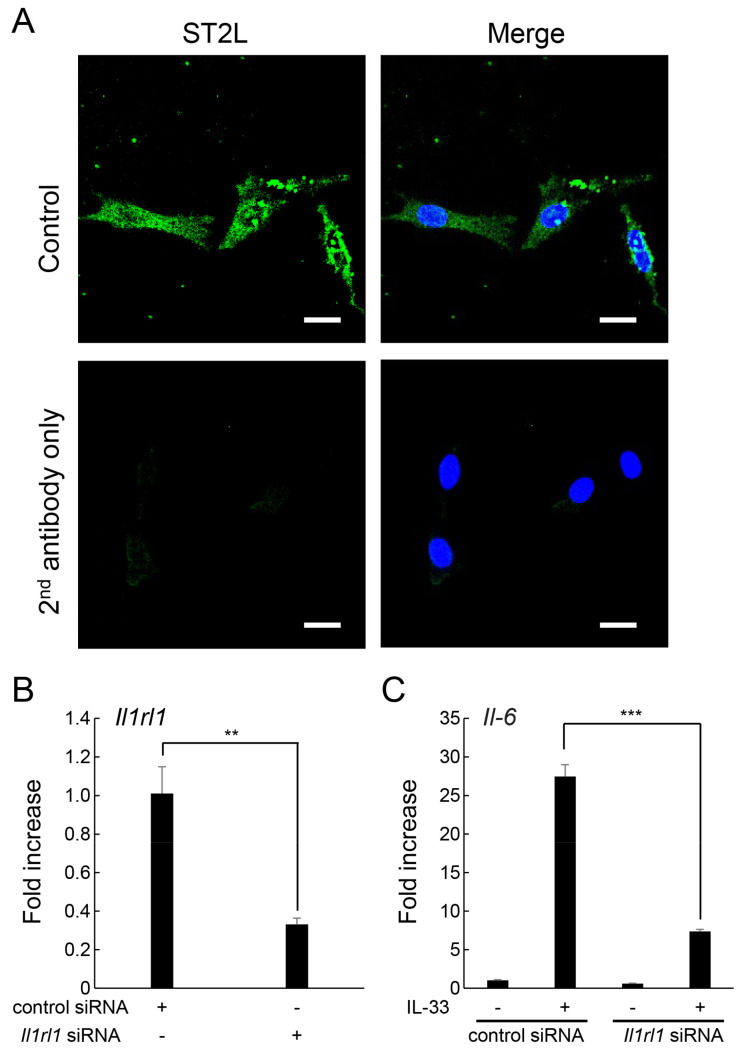
Involvement of ST2L in IL-33-induced increase in IL-6 expression. (**A**) MLO-Y4 cells were fixed and immune-stained with ST2L (green), and nuclei labeled with DAPI (blue). Cells stained without primary antibody were designated as negative control. Scale bars = 20 μm. (**B**) MLO-Y4 cells were transfected with control siRNA or *Il1rl1* siRNA and cultured for 48 h, and *Il1rl1* mRNA levels were measured using real-time RT-qPCR. ** *p* < 0.01 compared with ctr siRNA (Dunnett’s post hoc test after one-way ANOVA). (**C**) MLO-Y4 cells cultured as shown in (**B**) were incubated with IL-33 (10 µg/mL) for 6 h, and *Il-6* mRNA levels were measured using real-time RT-qPCR. *** *p* < 0.0001 compared with ctr siRNA stimulated with IL-33 (Tukey’s post hoc test after one-way ANOVA).

**Figure 3 ijms-24-14842-f003:**
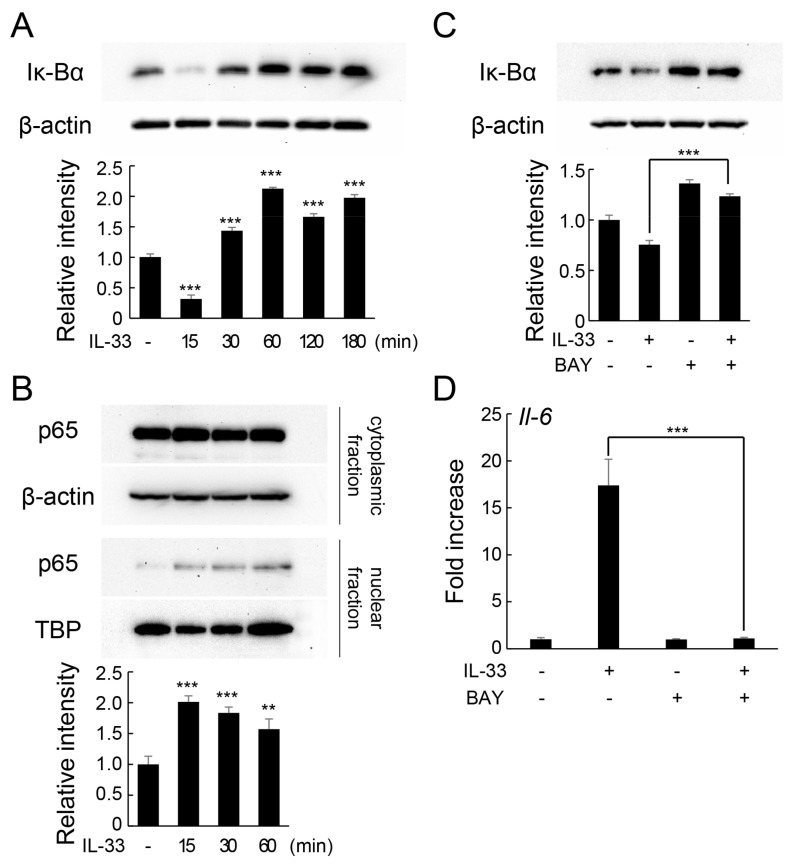
Involvement of NF-κB signaling in IL-33-induced increase in IL-6 expression in MLO-Y4 cells. (**A**) MLO-Y4 cells were incubated with IL-33 (10 ng/mL) for 15–180 min, and the protein levels of Iκ-Bα and β-actin were detected using Western blotting. Relative band intensities of Iκ-Bα protein normalized to changes in the β-actin protein were quantified by densitometric analyses. *** *p* < 0.001 compared with ctr (Dunnett’s post hoc test after one-way ANOVA). (**B**) MLO-Y4 cells were incubated with IL-33 (10 ng/mL) for 15–60 min. Cytoplasmic and nuclear fractions were prepared and analyzed for p65, β-actin, and TBP expression using Western blotting. Relative band intensities of p65 protein in nuclear fraction normalized to changes in the TBP protein were quantified by densitometric analyses. ** *p* < 0.01, *** *p* < 0.001 compared with ctr (Dunnett’s post hoc test after one-way ANOVA). (**C**) MLO-Y4 cells were pretreated with BAY11-7082 (BAY, 5 μM) for 1 h, followed by incubation with IL-33 (10 ng/mL) for 15 min. Protein levels of Iκ-Bα and β-actin were detected using western blot analysis. Relative band intensities of Iκ-Bα protein normalized to changes in the β-actin protein were quantified by densitometric analyses. *** *p* < 0.0001 compared with IL-33-stimulated cells (Tukey’s post hoc test after one-way ANOVA). (**D**) MLO-Y4 cells were pretreated with BAY11-7082 (BAY, 5 μM) for 1 h and incubated with IL-33 (10 ng/mL) for 6 h. *Il-6* mRNA levels were measured using real-time RT-qPCR. *** *p* < 0.0001 compared with IL-33-stimulated cells (Tukey’s post hoc test after one-way ANOVA).

**Figure 4 ijms-24-14842-f004:**
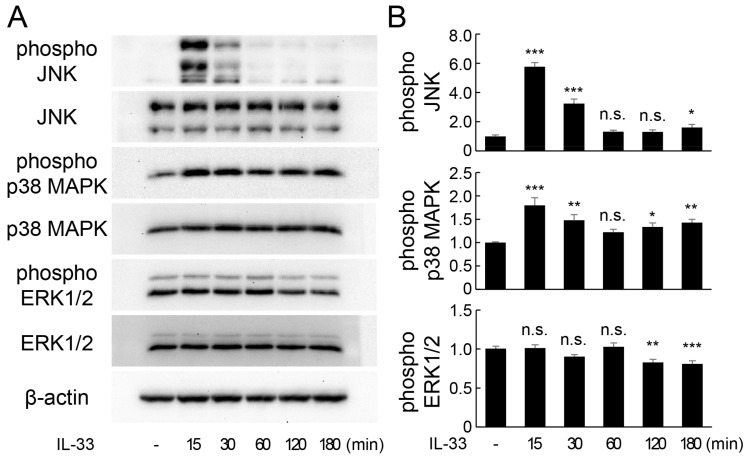
Activation of MAPK signaling by IL-33 in MLO-Y4 cells. (**A**) MLO-Y4 cells were incubated with IL-33 (10 ng/mL) for 15–180 min, and the protein levels of phospho-JNK, JNK, phospho-p38 MAPK, p38 MAPK, phospho-ERK1/2, ERK1/2, and β-actin were detected using Western blot analysis. (**B**) Relative band intensities of each phosphorylated protein normalized to changes in the total protein were quantified by densitometric analyses. * *p* < 0.05, ** *p* < 0.01, *** *p* < 0.001, n.s.; not significant compared with ctr (Dunnett’s post hoc test after one-way ANOVA).

**Figure 5 ijms-24-14842-f005:**
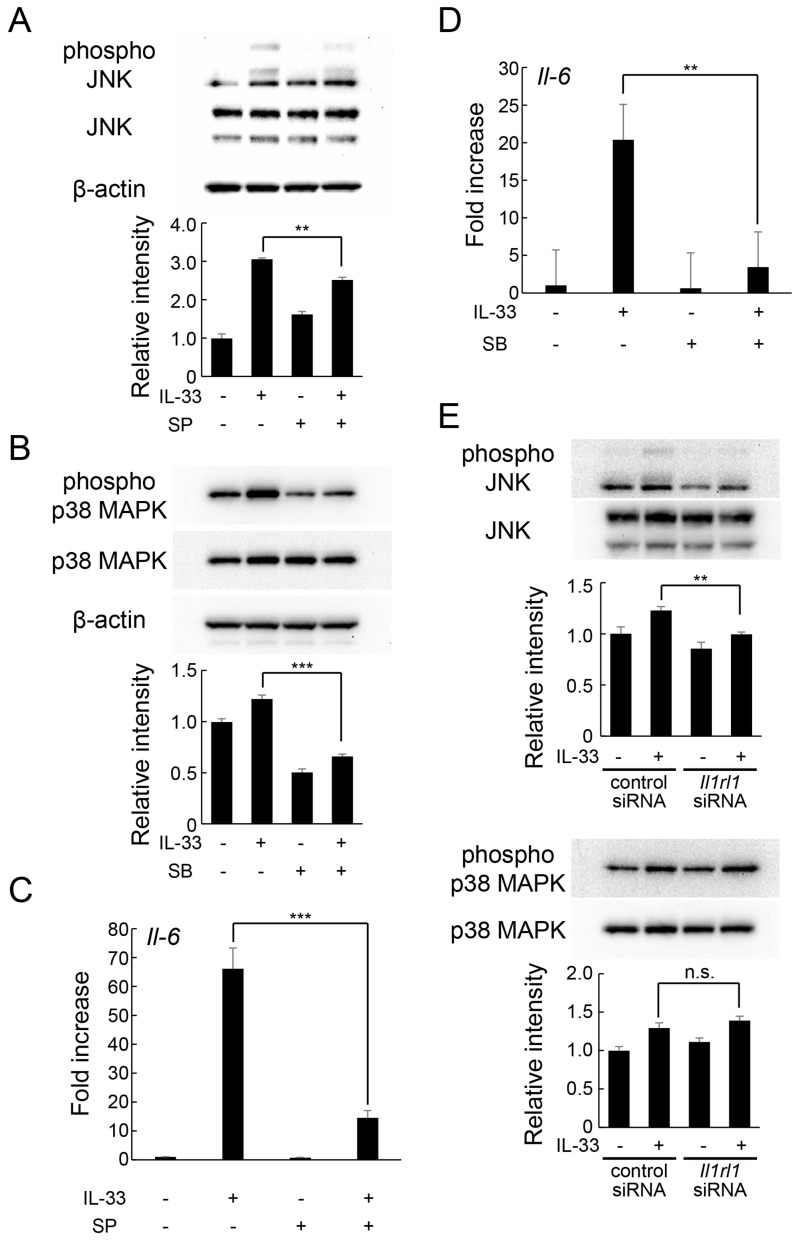
Involvement of MAPK signaling in IL-33-induced IL-6 expression in MLO-Y4 cells. (**A**,**B**) MLO-Y4 cells were pretreated with SP600125 (SP, 50 μM (**A**)) or SB203580 (SB, 10 μM (**B**)) for 1 h and incubated with IL-33 (10 ng/mL) for 15 min, and the protein levels of phospho-JNK, JNK, phosho-p38 MAPK, p38 MAPK, and β-actin were detected using Western blot analysis. Relative band intensities of each phosphorylated protein normalized to changes in the total protein were quantified by densitometric analyses. ** *p* < 0.01, *** *p* < 0.0001 compared with IL-33-stimulated cells (Tukey’s post hoc test after one-way ANOVA). (**C**,**D**) MLO-Y4 cells were pretreated with SP600125 (SP, 50 μM (**C**)) or SB203580 (SB, 10 μM (**D**) for 1 h and incubated with IL-33 (10 ng/mL) for 6 h. *Il-6* mRNA levels were measured using real-time RT-qPCR. ** *p* < 0.01, *** *p* < 0.0001 compared with IL-33-stimulated cells (Tukey’s post hoc test after one-way ANOVA). (**E**) MLO-Y4 cells were transfected with control siRNA or *Il1rl1* siRNA and cultured for 48 h, followed by incubation with IL-33 (10 µg/mL) for 6 h. The protein levels of phospho-JNK, JNK, phospho-p38 MAPK, p38MAPK, and β-actin were detected using Western blot analysis. Relative band intensities of each phosphorylated protein normalized to changes in the total protein were quantified by densitometric analyses. ** *p* < 0.01, n.s.; not significant compared with ctr siRNA stimulated with IL-33 (Tukey’s post hoc test after one-way ANOVA).

**Figure 6 ijms-24-14842-f006:**
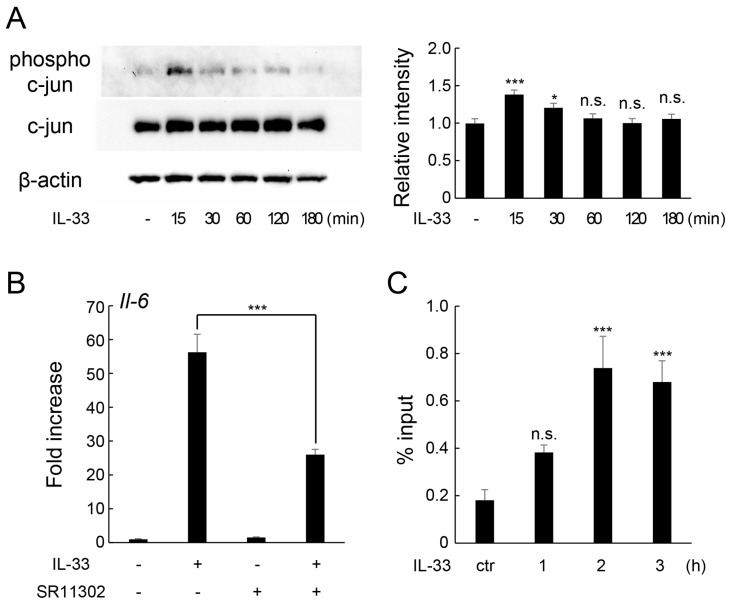
Involvement of c-Jun (AP-1) in IL-33-induced IL-6 expression in MLO-Y4 cells. (**A**) MLO-Y4 cells were incubated with IL-33 (10 ng/mL) for 15–180 min, and protein levels of phospho-c-Jun, c-Jun, and β-actin were detected using Western blot analysis. Relative band intensities of phospho-c-Jun protein normalized to changes in the c-Jun protein were quantified by densitometric analyses. * *p* < 0.05, *** *p* < 0.001, n.s.; not significant compared with ctr (Dunnett’s post hoc test after one-way ANOVA). (**B**) MLO-Y4 cells were pretreated with SR11302 (10 μM) for 30 min and incubated with IL-33 (10 ng/mL) for 6 h, and *Il-6* mRNA levels were measured using real-time RT-qPCR. *** *p* < 0.0001 compared with IL-33-stimulated cells (Tukey’s post hoc test after one-way ANOVA). (**C**) MLO-Y4 cells were incubated with IL-33 (10 ng/mL) for 1–3 h. Binding of c-Jun protein to the *Il-6* promoter region was confirmed using chromatin immunoprecipitation (ChIP) assay. Data represent the percentage input of independent triplicate samples. *** *p* < 0.001, n.s.; not significant compared with ctr (Dunnett’s post hoc test after one-way ANOVA).

**Table 1 ijms-24-14842-t001:** Real-time RT-qPCR primer sequences.

Gene	Primer Sequence (5′ - 3′)	
*Gapdh*	forward	5′- GAC GGC CGC ATC TTC TTG A -3′
	reverse	5′- CAC ACA CCG ACC TTC ACC ATT TT -3′
*Il-6*	forward	5′- GAG GAT ACC ACT CCC AAC AGA CC -3′
	reverse	5′- ATT GCT TGG GAT CCA CAC TCT CCA ACC TTT GAC -3′
*Tnf-α*	forward	5′- TCA TGC ACC ACC ATC AAG GA -3′
	reverse	5′- GAC ATT CGA GGC TCC AGT GAA -3′
*Il-1β*	forward	5′- AAG GGC TGC TTC CAA ACC TTT GAC -3′
	reverse	5′- TGG CGA GCT CAG GTA CTT CTG -3′
*Tnfsf11*	forward	5′- CTG ATG AAA GGA GGG AGC ACG -3′
	reverse	5′- GGA AGG GTT GGA CAC CTG AAT G -3′
*Opg*	forward	5′- GAG AGA AGC CAC GCA AAA GTG -3′
	reverse	5′- TCT TGG TAG GAA CAG CAA ACC TG -3′

## Data Availability

The data presented in this article are available upon request from the corresponding author.

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
