# Peer review of "The Mechanism of Interleukin 33-Induced Stimulation of Interleukin 6 in MLO-Y4 Cells"

_ijms, 2023, doi:10.3390/ijms241914842_

Round 1
Reviewer 1 Report
The study by Noguchi et al. aims to investigate the influence of IL-33 on the bone modulating functions of osteocyte-like MLO-Y4 cells in vitro. In particular, increased IL-6 expression through activation of the NF-κB, JNK/AP-1, and p38 MAPK signaling pathway is highlighted. The subject is very well introduced, the results are presented and discussed basically in a good way, nevertheless I have some points of criticism which I would like to elaborate in the following:
Major Issues:
1. The information that Tnfsf11 encodes RANKL should be mentioned either in parentheses in the introduction or in the presentation of results. Furthermore, the statistical information whether there is really a time- and dose-dependent increase of Il-6 is also missing here. As I understand it, the significance data are each related to the Ctrl condtion? Please make clear for each figure in the figure legends. There is a lack of information on how the conditions differ from each other. Please also indicate here and throughout the manuscript the corresponding statistical method in the figurelegends. If this is the same for all analyses, however, it is also sufficient to specify it in the method section.
2. It is not entirely clear what Il-1 rl-1 siRNA is directed against in the context of IL-33. Presumably against the receptor, but I did not find the abbreviation of the encoding gene in the corresponding section. This should be made up either in the introduction or in the presentation of results.
3. The presentation of the results of the signaling pathway analysis assumes that every reader knows the corresponding signaling pathways and what it means when corresponding proteins are phosphorylated or degraded. For the benefit of non-specialist readers, the results should at least briefly mention why exactly that protein is being investigated at this position.
4. The Western Blot results are currently only exemplary visualizations. The evaluations and statistical statements are missing. The authors should perform the corresponding evaluations of their WB (e.g. via ImageJ) and also show them in diagrams in the respective figures, including significance information. This assumes that the analyses were performed with the corresponding frequency as indicated by the method part. However, I do not find any information that the WB were evaluated semi/quantitatively at all. However, this must be included.
5. The discussion should have a part on the limitation of the study results and discuss critical points with regard to the experimental design (keyword in vitro). Furthermore, I see it as very critical to use only one reference gene for quantitative expression analysis (genes). This is no longer in line with the standard and should be urgently changed for future studies. Two reference genes are now considered the minimum for this type of analysis.
6. The authors should strongly focus their final statement on what they have shown. To what extent IL-33 really influences the functionality of osteocyte-like MLO-Y4 cells remains open. They have shown that IL-6 gene expression is affected and which signaling pathway IL-33 activates. Functionally, however, the authors did not include experimental approaches and analyses.
Minor Issues:
7. Please correct the figure legend 1C: MLO-Y4 were incubated ... for 12-48 h
8. Table 1: Gapdh should not be bold
9. The bars in the diagrams should always have the same width, regardless of how many bars are included (see Figure 1)
Author Response
Answers to Reviewer 1
The study by Noguchi et al. aims to investigate the influence of IL-33 on the bone modulating functions of osteocyte-like MLO-Y4 cells in vitro. In particular, increased IL-6 expression through activation of the NF-κB, JNK/AP-1, and p38 MAPK signaling pathway is highlighted. The subject is very well introduced, the results are presented and discussed basically in a good way, nevertheless I have some points of criticism which I would like to elaborate in the following:
We deeply appreciate your helpful and important suggestions.
Major Issues:
- The information that Tnfsf11 encodes RANKL should be mentioned either in parentheses in the introduction or in the presentation of results. Furthermore, the statistical information whether there is really a time- and dose-dependent increase of Il-6 is also missing here. As I understand it, the significance data are each related to the Ctrl condition? Please make clear for each figure in the figure legends. There is a lack of information on how the conditions differ from each other. Please also indicate here and throughout the manuscript the corresponding statistical method in the figure legends. If this is the same for all analyses, however, it is also sufficient to specify it in the method section.
Thank you for your helpful comments. We added the information of Tnfsf11 gene and modified the Results of the revised manuscript (lines 96-97) with a reference (No. 19). We also added the corresponding statistical method in the Figure legends.
- It is not entirely clear what Il1rl1 siRNA is directed against in the context of IL-33. Presumably against the receptor, but I did not find the abbreviation of the encoding gene in the corresponding section. This should be made up either in the introduction or in the presentation of results.
According to the reviewer’s comment, we added the information of Il1rl1 gene (encode ST2L) and modified the Results (lines 111-112) with a reference (No. 20).
- The presentation of the results of the signaling pathway analysis assumes that every reader knows the corresponding signaling pathways and what it means when corresponding proteins are phosphorylated or degraded. For the benefit of non-specialist readers, the results should at least briefly mention why exactly that protein is being investigated at this position.
Thank you very much for your important suggestions. We added the short information of each signaling pathway and modified the Results (lines 130-134 and lines 162-165) in the revised manuscript.
- The Western Blot results are currently only exemplary visualizations. The evaluations and statistical statements are missing. The authors should perform the corresponding evaluations of their WB (e.g. via ImageJ) and also show them in diagrams in the respective figures, including significance information. This assumes that the analyses were performed with the corresponding frequency as indicated by the method part. However, I do not find any information that the WB were evaluated semi/quantitatively at all. However, this must be included.
For the quantification of proteins, we performed the densitometric analyses on all the blots. As need for the space to represent the results of densitometric analyses, we divided first submitted Figure 4 into Figures 4 and 5 in the revised manuscript. We provided bar graphs and edited the Materials and Methods, Figure 3, 4, 5 and 6 in the revised manuscript.
- The discussion should have a part on the limitation of the study results and discuss critical points with regard to the experimental design (keyword in vitro). Furthermore, I see it as very critical to use only one reference gene for quantitative expression analysis (genes). This is no longer in line with the standard and should be urgently changed for future studies. Two reference genes are now considered the minimum for this type of analysis.
We appreciate your critical suggestions. According to the reviewer’s comments, we mentioned the limitation of this study and modified the Discussion (lines 331-336) in the revised manuscript.
- The authors should strongly focus their final statement on what they have shown. To what extent IL-33 really influences the functionality of osteocyte-like MLO-Y4 cells remains open. They have shown that IL-6 gene expression is affected and which signaling pathway IL-33 activates. Functionally, however, the authors did not include experimental approaches and analyses.
We appreciated your important comments. As noted by the reviewer, the present study mainly focused on the molecular mechanisms by which administration of IL-33 stimulates IL-6 expression in MLO-Y4 cells. However, the effects of IL-6 produced by MLO-Y4 cells on the differentiation and function are still unknown. We did not mention about it. So, we modified the Discussion (lines 331-336) in the revised manuscript.
Minor Issues:
- Please correct the figure legend 1C: MLO-Y4 were incubated ... for 12-48 h
Thank you for your suggestions. According to the reviewer suggestions, we corrected legend of Figure 1C.
- Table 1: Gapdh should not be bold
Thank you for your suggestions. We changed “Gapdh” to regular font in the Table 1.
- The bars in the diagrams should always have the same width, regardless of how many bars are included (see Figure 1)
According to the reviewer’s comment, we adjusted the widths of all bars on all graphs in the Figure to be the same. 

Reviewer 2 Report
Title: The Mechanism of Interleukin-33-induced Stimulation of Interleukin-6
in MLO-Y4 Cells.
In this paper the authors study osteocytes, IL-6 and IL-33 in periodontal tissues. The study aims to clarify the effect of IL-33 on the expression of regulatory factors for bone remodeling. An MLO-Y4 cell line is used in the in vitro model and the effect is analyzed using Western Blot analysis. IL-6 is analyzed after cellular treatment with IL-6. The authors conclude that the results suggest that IL-33 induces IL-6 expression and regulates osteocytes function through the activation of NF-κB, JNK/AP-1 and p38 MAPK pathways through interaction with receptors ST2L on the plasma membrane.
All in all, this paper seems good to me, however, I have some concerns that need to be clarified.
When addressing the problem of inflammation, the topic should be introduced by talking about immune sentinel cells that control inflammation and also tumors. Therefore, to make this paper more interesting for the readers of this important journal, the authors should expand a bit the introduction and discussion. Here, there is an article recently published that should be studied, incorporate the meaning and report in the list of references.
C. D’Ovidio. THE RESPONSE OF IMMUNE SENTINELS CAUSING INFLAMMATION IN GLIOMA AND GLIOBLASTOMA. European Journal of Neurodegenerative Diseases 2023; 12(2): 46-50. (www.biolife-publisher.it).
In addition, inflammatory cytokines and chemokines, including IL-1, IL-6, and tumor necrosis factor (TNF) mediate the immune response while insulin-like growth factors (IGFs), hormones that promote physiological growth, also participate in the inflammatory response . Here, there is an article recently published that should be studied, and incorporate the meaning.
This topic should be briefly mentioned in the discussion and reference list.
P. Conti and I. Tsilioni. IMPACT OF INSULIN-LIKE GROWTH FACTORS 1 AND 2 IN THE INFLAMMATORY RESPONSE MEDIATED BY CYTOKINES. European Journal of Neurodegenerative Diseases 2023; 12(1) : 5-10. (www.biolife-publisher.it).
- In the figures Il-1 and Il-6 should be IL-1 and IL-6.
- Fig.3 and 4 lacks a table quantifying the result.
I believe these suggestions are important for improving this paper. Without these corrections the paper cannot be published. So I recommend minor revision.
minor revision
Author Response
Answer to Reviewer 2
- In this paper the authors study osteocytes, IL-6 and IL-33 in periodontal tissues. The study aims to clarify the effect of IL-33 on the expression of regulatory factors for bone remodeling. An MLO-Y4 cell line is used in the in vitro model and the effect is analyzed using Western Blot analysis. IL-6 is analyzed after cellular treatment with IL-6. The authors conclude that the results suggest that IL-33 induces IL-6 expression and regulates osteocytes function through the activation of NF-κB, JNK/AP-1 and p38 MAPK pathways through interaction with receptors ST2L on the plasma membrane.
All in all, this paper seems good to me, however, I have some concerns that need to be clarified.
When addressing the problem of inflammation, the topic should be introduced by talking about immune sentinel cells that control inflammation and also tumors. Therefore, to make this paper more interesting for the readers of this important journal, the authors should expand a bit the introduction and discussion. Here, there is an article recently published that should be studied, incorporate the meaning and report in the list of references.
C. D’Ovidio. THE RESPONSE OF IMMUNE SENTINELS CAUSING INFLAMMATION IN GLIOMA AND GLIOBLASTOMA. European Journal of Neurodegenerative Diseases 2023; 12(2): 46-50. (www.biolife-publisher.it).
Thank you for your helpful comments We added suggested reference (No. 23) and modified the Discussion in the revised manuscript (lines 234-236).
- In addition, inflammatory cytokines and chemokines, including IL-1, IL-6, and tumor necrosis factor (TNF) mediate the immune response while insulin-like growth factors (IGFs), hormones that promote physiological growth, also participate in the inflammatory response. Here, there is an article recently published that should be studied, and incorporate the meaning.
This topic should be briefly mentioned in the discussion and reference list.
P. Conti and I. Tsilioni. IMPACT OF INSULIN-LIKE GROWTH FACTORS 1 AND 2 IN THE INFLAMMATORY RESPONSE MEDIATED BY CYTOKINES. European Journal of Neurodegenerative Diseases 2023; 12(1): 5-10. (www.biolife-publisher.it).
We appreciate your important suggestions. We added suggested reference (No. 25) and modified the Discussion in the revised manuscript (lines 242-244).
- In the figures Il-1 and Il-6 should be IL-1 and IL-6.
Thank you for your comments. According to the Rules for Nomenclature of Genes, Genetic Markers, Alleles, and Mutations in Mouse and Rat, gene symbols are italicized, with only the beginning of the word capitalized and the rest in lowercase. On the other hand, protein names are not italicized, but are all capitalized.
- 3 and 4 lacks a table quantifying the result.
For the quantification of proteins, we performed the densitometric analyses on all the blots. As need for the space to represent the results of densitometric analyses, we divided first submitted Figure 4 into Figures 4 and 5 in the revised manuscript. We provided bar graphs and edited the Materials and Methods, Figure 3, 4, 5 and 6 in the revised manuscript.
- I believe these suggestions are important for improving this paper. Without these corrections the paper cannot be published. So, I recommend minor revision.
Thank you very much for your helpful suggestions.
